# Anti-cancer effect of a novel photodynamic therapy using glucose-linked chlorin e6 conjugated trastuzumab for HER2-positive gastrointestinal cancers

Makiko Sasaki[1], Mamoru Tanaka[1*], Akihiro Nomoto[2], Ryusei Yamasaki[3], Tomokazu Yoshimura[4], Shigenobu Yano[5,6], Yasunari Sasaki[1], Yuki Kojima[1], Taketo Suzuki[7], Hirotada Nishie[7], Keiji Ozeki[1], Takaya Shimura[1], Eiji Kubota[1], Hiromi Kataoka[1]

**1** Department of Gastroenterology and Metabolism, Nagoya City University Graduate School of Medical Sciences, Aichi, Japan, **2** National Institute of Technology, Fukui College, Fukui, Japan, **3** Department of Applied Chemistry, Graduate School of Engineering, Osaka Metropolitan University, Osaka, Japan, **4** Department of Chemistry, Faculty of Science, Nara Women's University, Nara, Japan, **5** KYOUSEI Science Center for Life and Nature, Nara Women's University, Nara, Japan, **6** Department of Chemistry, Graduate School of Science, The University of Osaka, Osaka, Japan, **7** Department of Gastroenterology, Nagoya City University Midori Municipal Hospital, Aichi, Japan

\* mtanaka@med.nagoya-cu.ac.jp

## Abstract

Photodynamic therapy (PDT) is an anti-cancer therapy that employs a photosensitizer (PS) and an optimal wavelength of light, causing a photochemical reaction that releases reactive oxygen species, thereby inducing cancer cell death via oxidative stress. Because light irradiation is limited to the tumor site, PDT has minimal adverse effects. The cancer cell selectivity of the PS is important for reducing damage to the normal mucosa caused by scattered light. Antibody-drug conjugates (ADC) are novel anti-cancer therapies that combine a monoclonal tumor-surface-receptor-targeting antibody with a drug bonded through chemical linkers. ADCs enable the targeted delivery of a variety of drugs to cancer cells while minimizing their delivery to healthy tissues. One such tumor surface receptor is the human epidermal growth factor receptor 2 (HER2), which is of interest in the treatment of many cancers, including gastrointestinal cancer. To improve tumor selectivity and minimize damage to the mucosa surrounding the tumor in PDT, we established a novel PS glucose-linked chlorin e6-conjugated trastuzumab (G-Ce6-trastuzumab) that is conjugated to existing PS glucose-linked chlorin e6 (G-Ce6) and evaluated its anti-cancer effect compared to G-Ce6. The effect of PDT was evaluated using HER2-high-expression cells NCI-N87 and HER2-low-expression cells MKN-45. G-Ce6-trastuzumab is internalized by the intracellular organelles in cancer cells. Evaluation of cell death using the WST-8 assay also demonstrated a significantly higher cytotoxic effect of G-Ce6-trastuzumab in HER2-high-expression cells compared with conventional PS G-Ce6.

**Data availability statement:** All relevant data are within the manuscript.

**Funding:** This work was partially supported by the Japan Society for the Promotion of Science (JSPS) KAKENHI (grant number 22K20862) (to M.S.); JSPS KAKENHI (grant number 23K15019) (to M.S.); JSPS KAKENHI (grant number 25K19271) (to M.S.); Ichihara international Scholarship Foundation (grant number JOSE207012) (to M.S.); The Nitto Foundation (to M.S.); Nagoya Co-Creation Research Fund (to M.S.); JSPS KAKENHI (grant number 23K07358) (to M.T.); The Nitto Foundation (grant number 206093) (to M.T.); The foundation of Japan Cancer Society (grant number 207003) (to M.T.); The Hori sciences and arts foundation (grant number 207004) (to M.T.); The Murata Science Foundation (grant number 207024) (to M.T.); JSPS KAKENHI (grant number 24K08399) (to A.N.); JSPS KAKENHI (grant number 22K05144) (to S.Y.); JSPS KAKENHI (grant number 25K18881) (to Y.K.); JSPS KAKENHI (grant number 22K16050) (to H.N.); JSPS KAKENHI (grant number 20K16997) (to T.S.); JSPS KAKENHI (grant number 23K07421) (to H.K.); Grant-Aid for Outstanding Research Group Support Program in Nagoya City University (grant number 2401102) (to H.K.) and Grant-Aid for Promotion on Co-Creative Urban Development in Nagoya City University (grant number 2412143) (to H.K.). The funders had no role in the study design, data collection and analysis, decision to publish, or preparation of the manuscript.

**Competing interests:** The authors have declared that no competing interests exist.

Thereby, G-Ce6-trastuzumab may be an excellent novel PS for PDT because of its strong selectivity for HER2-high-expression cells.

---

## Introduction

The development of non-invasive treatment for cancers is particularly desirable in aging populations. Although cancer can occur at any age, most of cancers are generally considered to arise from the accumulation of genetic mutations caused by genetical and environmental factors. Consequently, with population aging, the absolute number of patients with cancer continues to increase worldwide [1]. Surgical resection and chemoradiotherapy, which are standard treatment options for cancer, are highly invasive and may not be feasible in elderly patients because of limited physiological reserve or poor general condition. Therefore, there is an increasing demand for less invasive anticancer therapies in aging societies. Photodynamic therapy (PDT), an anticancer treatment that utilizes photosensitizers (PSs) activated by light at optimal wavelengths and does not require surgery or systemic chemotherapy, has thus attracted considerable attention as a promising therapeutic option for elderly patients. Crucial components of PDT include light, oxygen, and PS. It is relatively non-invasive, as irradiation is limited to the cancer site, and PSs predominantly accumulate in cancer cells, thereby showing less systemic toxicity. However, mucosal damage to normal tissues owing to scattered light is unavoidable and remains a problem in PDT [2–5]. One approach to overcome this problem is to enhance the PS tumor selectivity. Tumor-specific accumulation of PSs is crucial for reducing tissue damage in the surrounding irradiated area and minimizing the required drug dosage. In recent years, novel antibody-drug conjugates (ADCs) have been developed and used in clinical practice to achieve tumor-specific drug delivery. ADCs are novel antibody-based drugs comprising monoclonal and cytotoxic antibodies. ADCs provide a unique opportunity to deliver drugs to tumor cells while minimizing their toxicity to normal tissues, achieving wider therapeutic windows and enhanced pharmacokinetic/pharmacodynamic properties. To date, 14 ADCs have been approved by the FDA, and more than 200 are under clinical development worldwide [6,7]. In this study, we developed a novel PS conjugated to a trastuzumab monoclonal antibody. Trastuzumab exerts its anticancer effects by specifically binding to human epidermal growth factor receptor 2 (HER2) protein, a gene product of the human oncogene HER2/neu (c-erbB-2). Trastuzumab was first introduced into clinical trials in the United States in 1992 and approved by the FDA in 1998 as the world's first humanized monoclonal antibody for the treatment of breast cancer. Currently, HER2-expressing advanced solid tumors are granted tumor-agnostic approval worldwide. The frequency of HER2 overexpression in gastric and gastroesophageal cancers ranged from 4.4% to 53.4%, with a mean of 17.9% [8–11]. Trastuzumab is an antibody-based drug with established clinical use in gastric cancer, making it a suitable candidate antibody component for the development of novel PS. We developed a novel PS glucose-linked chlorin e6 conjugated trastuzumab (G-Ce6-trastuzumab), by conjugating trastuzumab to our existing PS glucose-linked chlorin e6 (G-Ce6) and evaluated its anti-cancer effect compared with G-Ce6.

## Materials and methods

### Materials

The 31-(3-(1-thio-β-d-glucopyranosyl)propoxy) chlorin e6 trimethyl ester (G-Ce6) methyl (7S,8S)-18-ethyl-5-(2-methoxy-2-oxoethyl)-7-(3-methoxy-3-oxopropyl)-2,8,12,17-tetramethyl-13-(1-(3-(((2S,3R,4S,5S,6R)-3,4,5-trihydroxy-6-(hydroxy-methyl) tetrahydro-2H-pyran-2-yl)thio)propoxy)ethyl)-7H,8H-porphyrin-3-carboxylate (G-Ce6) was synthesized and provided by the laboratory of Fukui University (Japan). Trastuzumab (HY-P9907) was purchased from MedChem Express (Monmouth Junction, NJ, USA).

### Cell lines and culture

Human gastric cancer cell lines MKN-45 (No. 0254; Japanese Cancer Research Bank, Tokyo, Japan) and NCI-N87 (No.3814635; American Type Culture Collection, Manassas, VA, USA) were cultured in RPMI1640 (FUJIFILM Wako Pure Chemical Corporation, Osaka, Japan) supplemented with 10% fetal bovine serum and 1% ampicillin and strepto-mycin. Cells were cultured in an atmosphere with 5% carbon dioxide ($CO_2$) at 37 °C. All experiments in this study were performed using cells passaged fewer than 20 times after thawing.

### Flow cytometric analysis for PSs accumulation

Cells were seeded in 6-cm culture dishes and incubated for 24 h. Subsequently, the medium was replaced with fresh medium supplemented with PSs, and the cells were incubated with PSs for 0, 2, and 24 h to evaluate the accumulation of PSs. PSs were added to the culture medium at a final concentration of 9.6 nmol/L. After washing with PBS, cells were analyzed using a flow cytometer with excitation at 488 nm and emission at 695 nm. All flow cytometric examinations were performed in triplicate on a FACSCanto II (BD Biosciences, Ann Arbor, MI, USA), and 10,000 events were counted and analyzed using FlowJo software (BD Biosciences).

### *In vitro* PDT

Cells were incubated with various concentrations of PSs in the culture medium. The cells were then washed with PBS and irradiated with light-emitting diode light (CCS Inc., Kyoto, Japan) at 660 nm and 16 J/cm$^2$ (irradiance: 30.8 mW/cm2 × 520 s). After irradiation, the PBS in the wells was replaced with the culture medium, and the cells were incubated for the specified times before analysis.

### Reactive oxygen species assay

Intracellular reactive oxygen species (ROS) accumulation was determined using ROS detection reagents (Thermo Fisher Scientific). Cells were incubated with 9.6 n mol/L of PSs for 2 h and treated with PDT *in vitro*. Cells were then treated with H2DCF-DA (DCFH-DA, 2,'7'-dichlorodihydrofluorescein diacetate) at 10 μM in PBS for 15 min. Thereafter, the supernatant was replaced with fresh medium and the cells were incubated for 30 min. The increase in ROS levels was assessed by measuring the fluorescence of 5-(and-6)-carboxy-2′,7′-difluorodihydrofluorescein diacetate (carboxy-H2DFFDA) (492 nm/525 nm: excitation/emission) on a FACS Canto II. A minimum of 10,000 events was recorded for each sample.

### WST-8 assay

To determine cell survival rate at 24 h after PDT, the cells were incubated with a Cell Counting Kit-8 (CCK-8) reagents (Dojindo, Kumamoto, Japan) for 2 h according to the manufacturer's protocol, and the absorbance at 450 nm was measured using a microplate spectrophotometer (SPECTRA MAX340, Molecular Devices, Silicon Valley, CA, USA). Cell viability was expressed as a percentage of untreated control cells, and the half-maximal (50%) inhibitory concentration ($IC_{50}$) was calculated.

## Statistical analysis

Descriptive statistics and simple analyses were performed using Prism software (version 8.0; GraphPad Software, Inc., San Diego, CA, USA).

## Results

### Syntheses of G-Ce6-trastuzumab

Synthesis of G-Ce6-trastuzumab (Fig 1A) was conducted according to a previously reported DCC/sulfo-NHS method [12]. G-Ce6 (Fig 1B) was prepared according to a previous report [13] and hydrolysis of G-Ce6 yielded the corresponding carboxylic acid. In a 30 mL flask, G-Ce6 (500.7 mg, 0.560 mmol) was dissolved in methanol (50 mL). Aqueous $K_2CO_3$ (25 mL, 0.1 M) was added and the mixture was stirred overnight at room temperature. This mixture was acidified with 10 mL 10% hydrochloric acid and extracted with dichloromethane. Purification was carried out by column chromatography ($CH_2Cl_2$:MeOH = 10:1–5:1, twice) and checked by MALDI-TOF mass spectrometry (m/z: 878); hydrolyzed G-Ce6 was obtained in 26% yield.

In a 30 mL flask, hydrolyzed G-Ce6 (8.2 mg, 0.0093 mmol), DCC (2.9 mg, 0.014 mmol), and *N*-hydroxysulfosuccinimide (3.0 mg, 0.014 mmol) were dissolved in DMSO (1.8 mL), and this mixture was stirred overnight at room temperature. After the reaction, 0.1 eq. trastuzumab (solved in $H_2O$) was added and stirred overnight at room temperature. This mixture was concentrated (centrifugation for 40 min at 4,000 RCF) and washed with phosphate-buffered saline to obtain the final G-Ce6-trastuzumab product (Fig 1C). When using more than 0.1 eq. of trastuzumab, aggregation, or condensation proceeds to form many insoluble precipitates.

G-Ce6-trastuzumab levels were measured by ultraviolet-visible (UV-vis) spectroscopy in $H_2O$ (Fig 2). Remarkable absorption was observed at approximately 400 and 650 nm for chlorin e6. Peptides showed an absorbance of 1.0 at 280 nm at approximately 1 g/L, and the ε value of G-Ce6 was calculated as $1.75 \times 10^6$ L $mol^{-1}$ $cm^{-1}$ (ε654) thus, this resulting product consisted of $2.82 \times 10^{-7}$ mol/L of trastuzumab and $4.05 \times 10^{-8}$ mol/L of hydrolyzed G-Ce6.

### G-Ce6-trastuzumab showed higher accumulation in HER2-high-expression cells than in HER2-low-expression cells

We examined the accumulation of G-Ce6 and G-Ce6-trastuzumab *in vitro*. To compare the differences in accumulation according to the degree of HER2 expression in the cells, we examined the HER2-low expression gastric cancer cell line MKN-45 and the HER2-high-expression gastric cancer cell line NCI-N87. PS accumulation increased in a time-dependent manner in both cell lines. G-Ce6, which was not conjugated to trastuzumab, exhibited no difference in intracellular accumulation between MKN-45 and NCI-N87 cells (Fig 3A). Whereas G-ce6-trastuzumab accumulated higher amounts of NCI-N87 than MKN-45 cells (Fig 3B).

### Reactive oxygen species (ROS) production of G-Ce6-trastuzumab-PDT is significantly elevated in HER2-high-expression cells

We examined whether PDT using G-Ce6-trastuzumab could generate ROS and whether the therapeutic effect differed depending on the HER2 expression level in cells. The cells were stained with 5-(and-6)-carboxy-2',7'-difluorodihydrofluorescein diacetate (carboxy-H2DFFDA), which detects ROS, including singlet oxygen, superoxide, hydroxyl radicals, peroxide, and hydroperoxide. No differences in ROS production were observed between the two cell lines with different HER2 expression levels during PDT with G-Ce6. In contrast, PDT with G-Ce6-trastuzumab showed significantly stronger ROS production in NCI-N87 cells than in MKN45 cells (Fig 4).

**Fig 1. Chemical structures of photosensitizers (PSs). (A)** Structure of G-Ce6–trastuzumab. **(B)** Structure of G-Ce6; methyl (7*S*,8*S*) −18-ethyl-5- (2-methoxy-2-oxoethyl) −7- (3-methoxy-3-oxopropyl) −2,8,12,17-tetramethyl-13- (1- (3- (((2*S*,3*R*,4*S*,5*S*,6*R*) −3,4,5-trihydroxy-6- (hydroxymethyl) tetrahydro-2*H*-pyran-2-yl) thio) propoxy) ethyl) -7*H*,8*H*-porphyrin-3-carboxylate (G-Ce6). **(C)** Schematic diagram of the synthetic process from G-Ce6 to G-ce6–trastuzumab.

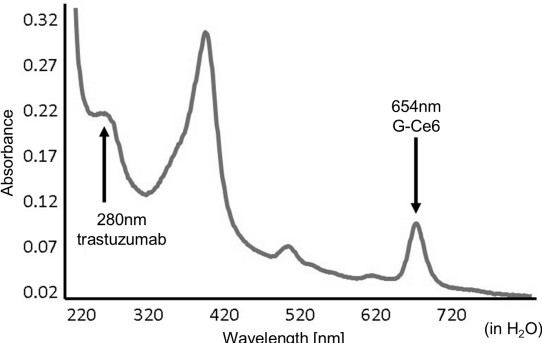

**Fig 2. UV-vis. Spectrum of G-Ce6–trastuzumab.** Remarkable absorption was observed at approximately 400 and 650 nm for chlorin e6. Peptides show an absorbance of 1.0 at 280 nm as almost 1 g/L, and the ε value of G-Ce6 was calculated.

### PDT with G-Ce6-trastuzumab induces a significant cell-killing effect in HER2- high-expression cells

To determine the $IC_{50}$ 24 h after irradiation, a WST-8 assay was performed. As shown in Fig 5, PDT with G-Ce6-trastuzumab induced a dose-dependent cell-killing effect. No difference in cell-killing effect was observed between the two cell lines with different HER2 expression levels during PDT using G-Ce6 ($IC_{50}$ MKN-45 vs. NCI-N87; 5.98 [± 1.30] vs. 8.53 [± 2.17] nmol/L). In contrast, PDT using G-Ce6-trastuzumab showed a stronger therapeutic effect in HER2-high-expression NCI-N87 compared with HER2-low-expression MKN-45 ($IC_{50}$ MKN-45 vs. NCI-N87; 3.16 [± 0.68] vs. 1.14 [± 0.40] nmol/L).

### Discussion

PDT is a relatively non-invasive anti-cancer therapy that employs a PS with an optimal wavelength of light irradiation. Several researchers have found that the combination of light with certain chemicals can induce cell death. The modern era of PDT began in the 1960s when Lipson reported a hematoporphyrin derivative as a PS [14,15]. Since then, many PSs have been developed. To increase the anti-tumor effects and cancer cell accumulation, we developed a novel PS for PDT, G-Ce6 [13,16]. Generally, cancer cells take up higher levels of glucose than normal cells, a phenomenon known as the Warburg effect [17], which is also utilized in positron emission tomography (PET-CT). Based on this effect, we synthesized more than 30 types of sugar-conjugated chlorins [18–27] resulting in G-Ce6, which showed the strongest antitumor effects in PDT ever reported for synthesized sugar-conjugated chlorins [28–30]. Furthermore, G-Ce6 was rapidly discharged from the body. Even with such an excellent PS, damage to the surrounding mucosa caused by scattered light cannot be ignored. By enhancing the selectivity of PS in tumors, the relative accumulation in normal tissues surrounding the tumor can be reduced, thereby minimizing the damage to normal tissues caused by PDT. Therefore, a higher degree of tumor accumulation is required for PSs.

To enhance the cancer cell selectivity of PS, we focused on the HER2 antibody drug trastuzumab, which has been used clinically to treat many cancers, including gastric cancer. The HER2 protein is a 185 kDa transmembrane receptor that belongs to the tyrosine kinase epidermal growth factor receptor family, which promotes cell growth, division, and motility. HER2 overexpression occurs in approximately 20% of patients with breast cancer and is associated with poor outcomes [31]. Compared to other subtypes, HER2-positive cancers grow faster because of increased HER2 signaling. HER2-targeted molecular therapies have become the standard treatment for HER2-positive cancers and have demonstrated excellent clinical efficacy. Considering that HER2 is expressed in various types of cancers, PDT using G-Ce6-trastuzumab has the potential for clinical application in all cancers expressing HER2.

PDT using G-Ce6-trastuzumab efficiently accumulated in the cells, and a clear difference was observed between HER2-low-expressing MKN-45 and HER2-high-expressing NCI-N87 cells (Fig 2). Consistent with this difference in cellular

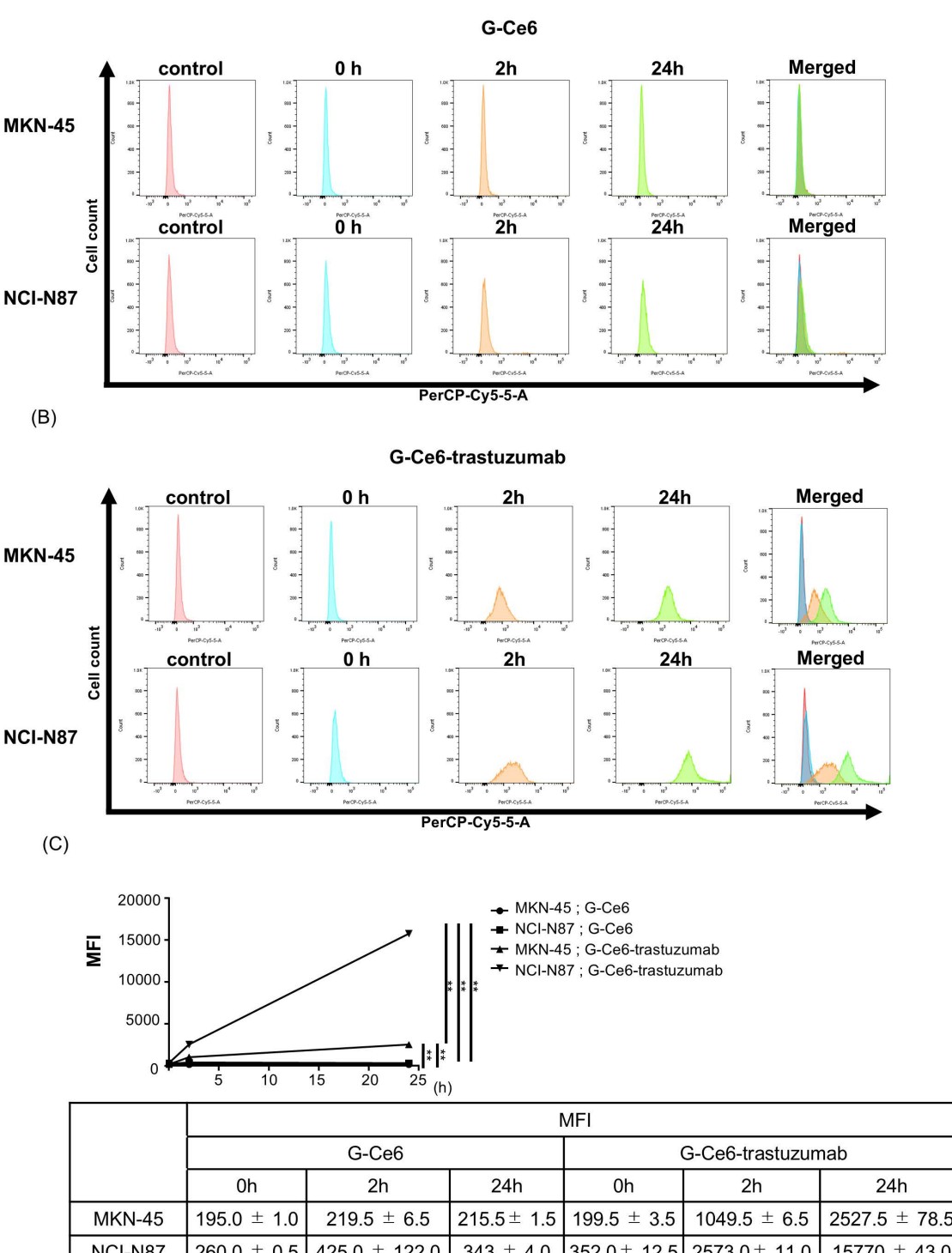

**Fig 3. Flow cytometric analysis for PS accumulation.** Histograms of **(a)** G-Ce6 and **(b)** G-Ce6-trastuzumab from flow cytometric analysis. **(c)** Temporal changes in mean fluorescence intensity (MFI). The abscissa indicates the intensity of the emission and the ordinate represents the number of cells. Significance was determined using Tukey's multiple comparison test and was set at ** P< 0.01.

(A)

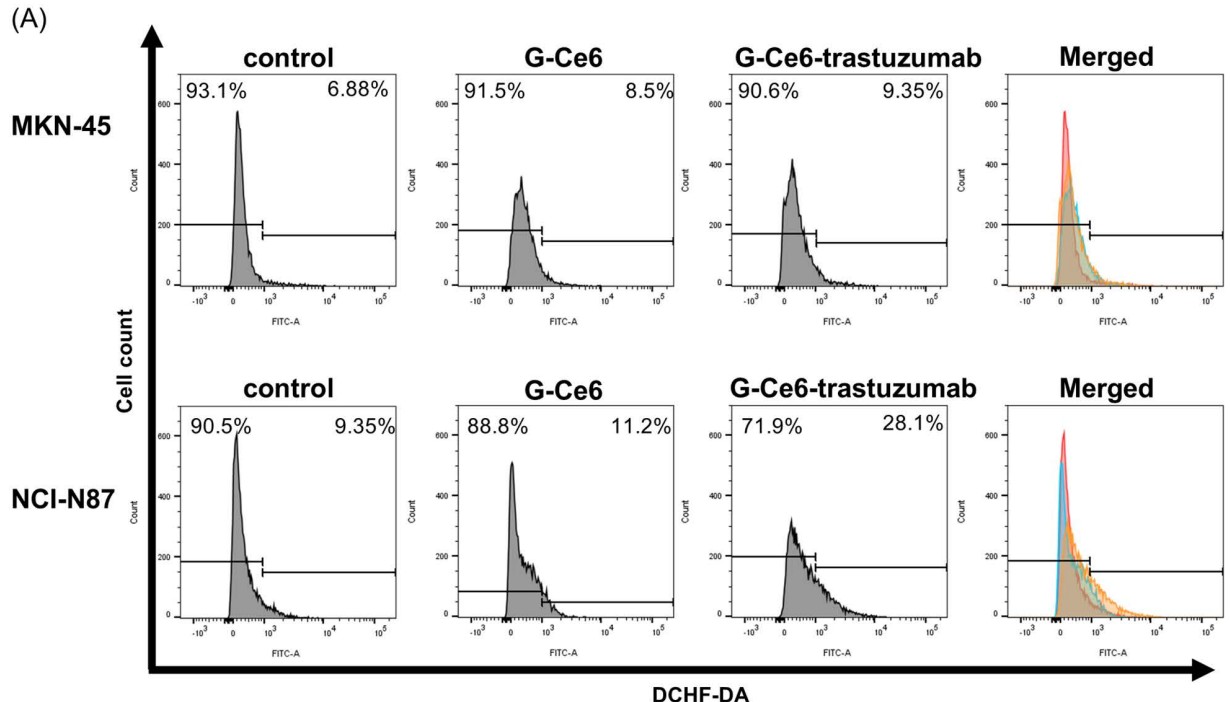

(B)

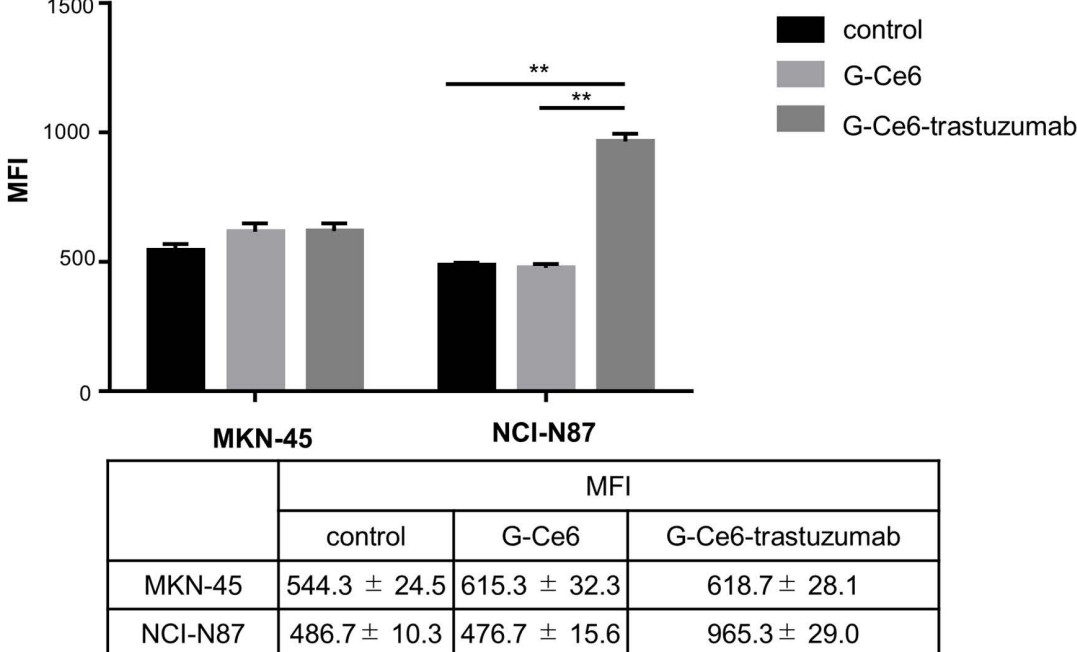

| | MFI | | |
|---|---|---|---|
| | control | G-Ce6 | G-Ce6-trastuzumab |
| MKN-45 | 544.3 ± 24.5 | 615.3 ± 32.3 | 618.7 ± 28.1 |
| NCI-N87 | 486.7 ± 10.3 | 476.7 ± 15.6 | 965.3 ± 29.0 |

**Fig 4. Reactive oxygen species (ROS) induction by PDT with PSs.** Changes in ROS generation induced by PDT with G-Ce6 or G-Ce6-trastuzumab according to HER2 expression levels. **(a)** Histograms and **(b)** MFI changes of ROS generation in MKN45 and NCI-N87 cells. Significance was determined using Tukey's multiple comparison test and was set at ** $P < 0.01$.

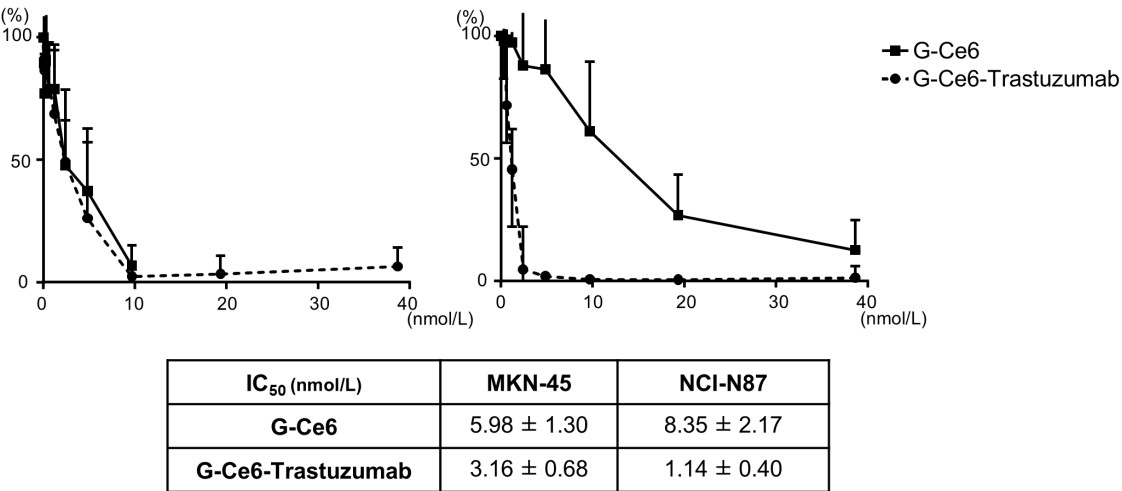

| IC$_{50}$ (nmol/L) | MKN-45 | NCI-N87 |
|---|---|---|
| G-Ce6 | 5.98 ± 1.30 | 8.35 ± 2.17 |
| G-Ce6-Trastuzumab | 3.16 ± 0.68 | 1.14 ± 0.40 |

**Fig 5. WST-8 assay of cells treated by PDT with PSs.** Cell survival rate at 24 h after PDT was evaluated using the WST-8 assay and expressed as the 50% inhibitory concentration (IC$_{50}$). The cells were incubated with various concentrations of PSs and irradiated with 660 nm LED light at 16 J/cm$^2$.

uptake, the cytotoxic effects were significant (Fig 4 and 5). Antibody-conjugated PSs, by exploiting the specificity of tumor-targeting antibodies, enable a reduction in the PS dose required to achieve therapeutic efficacy. Consequently, this strategy is expected to decrease unnecessary systemic accumulation of PS and, in clinical practice, shorten the mandatory light-shielding period following treatment, thereby improving patient safety and treatment feasibility.

This study has several limitations. In particular, the ROS detection method used in this study mainly detects relatively long-lived ROS (e.g., peroxides) and therefore does not comprehensively reflect all ROS species generated by PDT. DCFH-DA needs to be present during irradiation. However, the concentration of the photosensitizer used in this study (9.6 nmol/L) was relatively low, and we were concerned that prolonged incubation under PBS conditions containing H2DCFDA prior to irradiation could further dilute or promote leakage of the photosensitizer from the cells. Therefore, we administered H2DCFDA immediately after irradiation and evaluated ROS generation under these conditions. Another limitation of this study is that slower mechanisms cell death, such as autophagy [32], were not investigated. Nevertheless, because the PDT protocol was designed as a single light irradiation and cytotoxic effects are generally apparent within 24 hours as acute photokilling, the CCK-8 assay, that uses WST-8 and can detect acute cell damage more sensitively than the MTT assay, was considered appropriate for evaluating acute cell death in this study.

In conclusion, our results indicate that G-Ce6-trastuzumab may be an excellent antibody-conjugated novel PS for PDT because of its strong anti-cancer effects.

## Acknowledgments

We thank Akiko Yamato and Yukimi Ito for their technical assistance and the Research Equipment Sharing Center of Nagoya City University. This work conducted in Nara Institute of Science and Technology was supported by "Advanced Research Infrastructure for Materials and Nanotechnology in Japan (ARIM)" of the Ministry of Education, Culture, Sports, Science and Technology (MEXT).

## Author contributions

**Conceptualization:** Mamoru Tanaka, Akihiro Nomoto.

**Data curation:** Makiko Sasaki, Mamoru Tanaka, Akihiro Nomoto, Ryusei Yamasaki, Yasunari Sasaki, Yuki Kojima, Taketo Suzuki, Hirotada Nishie, Keiji Ozeki, Takaya Shimura, Eiji Kubota.

**Formal analysis:** Makiko Sasaki, Akihiro Nomoto.

**Funding acquisition:** Makiko Sasaki, Mamoru Tanaka, Akihiro Nomoto, Shigenobu Yano, Yuki Kojima, Taketo Suzuki, Hirotada Nishie.

**Investigation:** Makiko Sasaki, Mamoru Tanaka, Akihiro Nomoto, Ryusei Yamasaki.

**Methodology:** Makiko Sasaki, Akihiro Nomoto.

**Project administration:** Makiko Sasaki.

**Resources:** Makiko Sasaki, Tomokazu Yoshimura, Shigenobu Yano.

**Software:** Makiko Sasaki.

**Supervision:** Mamoru Tanaka, Tomokazu Yoshimura, Hiromi Kataoka.

**Validation:** Mamoru Tanaka, Hiromi Kataoka.

**Writing – original draft:** Makiko Sasaki.

**Writing – review & editing:** Mamoru Tanaka, Hiromi Kataoka.

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
