## [Decision Letter · Decision Letter 0]

9 Dec 2025

Dear Dr. Tanaka,

Thank you for submitting your manuscript to PLOS ONE. After careful consideration, we feel that it has merit but does not fully meet PLOS ONE’s publication criteria as it currently stands. Therefore, we invite you to submit a revised version of the manuscript that addresses the points raised during the review process.

We look forward to receiving your revised manuscript.

Kind regards,

Michael R Hamblin

Academic Editor

PLOS One

Journal Requirements:

https://www.mdpi.com/2072-6694/11/5/636

https://www.mdpi.com/1422-0067/25/6/3233

In your revision ensure you cite all your sources (including your own works), and quote or rephrase any duplicated text outside the methods section. Further consideration is dependent on these concerns being addressed.

Additional Editor Comments:

The reviewer has pointed out some deficiencies in the methodology that must be corrected. Clonogenic assay, improved ROS detection assays (both before and after light), justification for the high in vitro light dose.

Reviewer's Responses to Questions

**Comments to the Author**

1. Is the manuscript technically sound, and do the data support the conclusions?

Reviewer #1: Partly

2. Has the statistical analysis been performed appropriately and rigorously?

Reviewer #1: Yes

3. Have the authors made all data underlying the findings in their manuscript fully available?

Reviewer #1: Yes

4. Is the manuscript presented in an intelligible fashion and written in standard English?

Reviewer #1: Yes

Reviewer #1: PONE D-25 Nov 6

This report relates to the use of a photosensitizing agent for treatment of GI cancer. Line 34: if the selectivity of the photosensitizer is appropriate, there should be no damage to ‘normal mucosa’ because these cells should not be photosensitized. If there is no selectivity for sites of neoplasia, a CO2 laser could be used for therapy. It appears that the agent being used is thought to have insufficient targeting properties so it need to be conjugated to an antibody. There are many photosensitizing agents in use today that do not need to be conjugated to anything.

In this report, chlorin e6 is conjugated to trastuzumab. Glucose is involved. Cell death is evaluated using ‘the WST-8 assay’. It is claimed that photosensitizer specificity needs to be improved to avoid adverse effects to the patient. In vitro studies involved a 16 J/sq cm light dose which is very high for in vitro experiments. For most such studies, light doses in the range of 300-500 mJ/sq cm are used.

Details for the ROS assay indicate (line 120) that cells were irradiated, then exposed to DCFH-DA which is incorrect. DCFH-DA needs to be present during irradiation, since the ROS being formed are rapidly quenched. This error indicates that the authors are inexperienced in PDT technology. What is the basis for the cell counting kit as an indicator of photokilling? The most unambiguous approach is to use a clonogenic assay which measures the ability of cells to form colonies. Anything else is only an approximation. Oleinick described a comparison between clonogenic assays and an alternative approach (Photochem Photobiol 2009) and found that the correlation was poor. Line 232 indicates a ‘specific wavelength of light’. All wavelengths of light are ‘specific’. What is needed is light at a wavelength that corresponds to an absorbance optimum of the photosensitizer, preferably one that can penetrate into tissues.

Summary: The ROS detection procedure should be carried out with the reagent present during irradiation. Any result that claims to detect photokilling should involve a clonogenic assay which detects the ability of cells to proliferate. It appears that the photosensitizer being used requires the presence of an antibody to provide adequate targeting. There are photosensitizers that do not have such a requirement. Does this agent represent a significant improvement over what is already available? It is true that the use of 654 nm light would represent an improvement over the 630 nm wavelength used with Photofrin.

.

Reviewer #1: No

---

## [Author Response · Author response to Decision Letter 1]

20 Dec 2025

Date: 09-Dec-2025

PONE-D-25-57729

Title: Anti-cancer effect of a novel photodynamic therapy using glucose-linked chlorin e6 conjugated trastuzumab for HER2-positive gastrointestinal cancers

Major revision

Journal Requirements:

I revised my manuscript, and confirmed that my revised manuscript meets PLOS ONE's style requirements.

I cited our research group’s report (https://www.mdpi.com/2072-6694/11/5/636) to manuscript that was previously published. However, I can’t point out the overlapping text with the publication https://www.mdpi.com/1422-0067/25/6/3233. Although there may be some overlap in the methodologies, it is minimal.

I have no Supporting Information file.

There is no recommendation to cite specific previously published works.

Reviewer’s Comments to Author:

Reviewer: 1

This report relates to the use of a photosensitizing agent for treatment of GI cancer.

1) Line 34: if the selectivity of the photosensitizer is appropriate, there should be no damage to ‘normal mucosa’ because these cells should not be photosensitized.

Thank you for your comment. To the best of our knowledge, there is no photosensitizer that shows absolute and exclusive accumulation in tumor cells without any distribution to normal cells. Furthermore, while it is theoretically correct that super selective photosensitizers should not damage normal mucosa, drug distribution can be affected by inflammation, regenerative epithelium, and tumor heterogeneity in clinical cases. Therefore, improving selectivity remains an important clinical challenge. Tumor selectivity in photodynamic therapy remains a relative concept, and strategies to further enhance tumor-specific accumulation are still required.

2) If there is no selectivity for sites of neoplasia, a CO2 laser could be used for therapy.

Thank you for your comment. While CO₂ laser ablation may be feasible for superficial tumor removal, the extremely shallow penetration depth limits its usefulness for reducing advanced tumor volume in gastrointestinal cancers. Therefore, CO₂ laser therapy aimed at gastrointestinal tumor debulking is not commonly adopted in current clinical practice. In contrast, photodynamic approaches offer the potential to affect tumor tissue beyond the immediate surface.

3) It appears that the agent being used is thought to have insufficient targeting properties so it need to be conjugated to an antibody. There are many photosensitizing agents in use today that do not need to be conjugated to anything.

We appreciate your comment. Talaporfin sodium (TS) is currently the most widely used photosensitizer in clinical practice in Japan. Compared with the conventional agent porfimer sodium, TS exhibits enhanced tumor accumulation and a substantially shorter light-shielding period, which has led to its widespread clinical adoption in Japan. G-Ce6 is a novel photosensitizer that we synthesized and reported in 2018. In vitro photodynamic therapy using G-Ce6 demonstrated an anti-tumor effect that was approximately 9,000–34,000 times stronger than that of TS (These points were described in Reference 8. In addition, we have added an additional reference 11 (Osaki et al, Cancers, 2019.) related to G-Ce6 in the revised manuscript.). Taken together, these findings indicate that G-Ce6 is a highly potent photosensitizer with markedly enhanced cellular accumulation and photodynamic efficacy. The rationale for using an antibody-conjugated photosensitizer in this study was to introduce molecular targeting toward HER2-positive tumors. Our approach is intended as a complementary strategy for HER2-positive tumors, rather than a universal replacement for existing agents. Enhanced tumor accumulation compared with conventional agents provides several clinical advantages, including not only a reduction in damage to surrounding normal mucosa caused by scattered light, but also the feasibility of a shorter light-shielding period.

4) In vitro studies involved a 16 J/sq cm light dose which is very high for in vitro experiments. For most such studies, light doses in the range of 300-500 mJ/sq cm are used.

Thank you for your comment. We reviewed several publications on photodynamic therapy (PDT) using TS and found that, in in vitro experiments, irradiation is generally performed at the same order of magnitude as in our study, and none of the studies used irradiation doses in the mJ/cm² range. (7.5 J/cm2; Ezzaldeen et al, 2025, 10 J/cm2; Miki et al, 2013, 13 J/cm2; Akter et al, 2019, 0～10 J/cm2; Ichikawa et al, 2013). Therefore, we believe that this irradiation dose cannot be considered outside the standard range.

5) Details for the ROS assay indicate (line 120) that cells were irradiated, then exposed to DCFH-DA which is incorrect. DCFH-DA needs to be present during irradiation, since the ROS being formed are rapidly quenched. This error indicates that the authors are inexperienced in PDT technology.

Thank you for your comment. According to the protocol for H2DCFDA that we used (Invitrogen™), the recommended incubation time is 5–60 minutes. It is well known that it may cause dye–dye quenching when H2DCFDA is present in excess within cells. As you pointed out, we performed irradiation after co-incubating both the photosensitizers (G-Ce6 and G-Ce6-trastuzumab) and H2DCFDA for 60 minutes so that H2DCFDA would be present during irradiation; however, this approach did not yield satisfactory results. We consider that this may be due to the 60-minute incubation being insufficient for adequate cellular uptake of the photosensitizers.

An alternative approach—incubating cells with the photosensitizer first, washing, and then adding H2DCFDA in PBS before irradiation—was not performed, because the concentration of the photosensitizer used in this study (9.6n mol/L ; we apologize for the error in the initial version and have corrected it) was relatively low, and changing to PBS and incubating for longer time under PBS before irradiation was a concern as it could further dilute the photosensitizers.

Therefore, consistent with several previous reports (Huang et al, BMC cancer, 2019; Fatima et al, Journal of Advanced Research 2021), we administered H2DCFDA immediately after irradiation and evaluated ROS generation under these conditions.

6) What is the basis for the cell counting kit as an indicator of photokilling? The most unambiguous approach is to use a clonogenic assay which measures the ability of cells to form colonies. Anything else is only an approximation. Oleinick described a comparison between clonogenic assays and an alternative approach (Photochem Photobiol 2009) and found that the correlation was poor.

Thank you for your comment. We acknowledge that clonogenic assays are the gold standard for evaluating long-term reproductive cell death and are particularly appropriate for assessing cytotoxicity after prolonged drug exposure.

In this study, however, PDT involves a single light irradiation, and cytotoxic effects are typically evident within 24 hours after treatment. Therefore, for evaluating acute photokilling following a single PDT session, we consider the cell counting kit assay to be appropriate.

In contrast, clonogenic assays require long-term culture, during which additional confounding factors may influence colony formation. While clonogenic assays are valuable for long-term survival analysis, we believe that the cell counting kit assay is suitable for the aims of the present study.

7) Line 232 indicates a ‘specific wavelength of light’. All wavelengths of light are ‘specific’. What is needed is light at a wavelength that corresponds to an absorbance optimum of the photosensitizer, preferably one that can penetrate into tissues.

Thank you for your comment. We acknowledge that the term “specific” may cause misunderstanding. Accordingly, we have revised the expression to “optimal for the photosensitizer” to more accurately reflect our intention.

Summary: The ROS detection procedure should be carried out with the reagent present during irradiation. Any result that claims to detect photokilling should involve a clonogenic assay which detects the ability of cells to proliferate. It appears that the photosensitizer being used requires the presence of an antibody to provide adequate targeting. There are photosensitizers that do not have such a requirement. Does this agent represent a significant improvement over what is already available? It is true that the use of 654 nm light would represent an improvement over the 630 nm wavelength used with Photofrin.

Thank you for your valuable comments. We acknowledge that the presence of the ROS detection reagent during light irradiation is an important consideration. However, as described above, due to issues related to the concentration of the photosensitizer and the required incubation time, we addressed this concern by adding the ROS detection reagent H2DCFDA immediately after irradiation. In this study, our primary objective was to evaluate acute photokilling following a single PDT session using the cell counting kit assay. The newly developed photosensitizer G-Ce6-trastuzumab is designed to achieve high tumor accumulation through antibody-mediated targeting. The key features of this photosensitizer include the potential to reduce phototoxicity in non-tumor tissues by improving tumor specificity, as well as a reduction in the required drug dose. We consider these points to represent important advantages of this agent over existing photosensitizers.

Furthermore, the ability to use long-wavelength light at 654 nm offers a clear advantage in terms of improved tissue penetration compared with the 630 nm wavelength used for Photofrin, which we believe is of clinical relevance.

Taken together, compared with conventional photosensitizers, this novel photosensitizer G-Ce6-trastuzumab demonstrates clinical significance in terms of (i) enhanced tumor selectivity through targeting, (ii) reduced nonspecific phototoxicity, and (iii) improved tissue penetration enabled by long-wavelength excitation.

---

## [Decision Letter · Decision Letter 1]

30 Dec 2025

Dear Dr. Tanaka,

Thank you for submitting your manuscript to PLOS ONE. After careful consideration, we feel that it has merit but does not fully meet PLOS ONE’s publication criteria as it currently stands. Therefore, we invite you to submit a revised version of the manuscript that addresses the points raised during the review process.

We look forward to receiving your revised manuscript.

Kind regards,

Michael R Hamblin

Academic Editor

PLOS One

**Journal Requirements:**

Additional Editor Comments:

The reviewer still has concerns. You should use both PBS and medium for PDT, measure ROS both during and after light and not use "viability" for CC8-K assay.

Reviewers' comments:

Reviewer's Responses to Questions

**Comments to the Author**

Reviewer #1: (No Response)

2. Is the manuscript technically sound, and do the data support the conclusions?

Reviewer #1: Partly

3. Has the statistical analysis been performed appropriately and rigorously?

Reviewer #1: N/A

4. Have the authors made all data underlying the findings in their manuscript fully available?

Reviewer #1: No

5. Is the manuscript presented in an intelligible fashion and written in standard English?

Reviewer #1: No

Reviewer #1: Looking sequentially through this report, why is ‘the aging population’ singled out (line 53)? Cancer can occur at any age. Where is the data indicating ‘mucosal damage’ (line 57)? The light dose needed for efficacy (line 113) is substantially higher than most reported uses of PDT in vitro. Cells in PBS are in a non-physiologic state which can initiate death pathways. Why was culture medium not used?

As pointed out before, ROS do not persist for any significant time after formation. This means that DCFH-DA needs to be present during irradiation, not after. Only a few long-lasting ROS, e.g., peroxides, will be present when irradiation ends. The accurate method for assessing viability is via clonogenic assays. If the authors want to use the CCK-8 kit, they need to report exactly what this kit measures, and it isn’t viability (as Oleinick showed in 2009).

Summary: lacking an appropriate assay for ROS formation, there is no accurate estimate of formation of these species. The word ‘viability’ is incorrectly used in several places. With no pertinent assay for viability (clonogenics), this term cannot be used. The authors could report what they have found but need to be accurate in claims for what the results indicate.

.

Reviewer #1: No

---

## [Author Response · Author response to Decision Letter 2]

10 Jan 2026

Date: 10-Jan-2026

PONE-D-25-57729

Title: Anti-cancer effect of a novel photodynamic therapy using glucose-linked chlorin e6 conjugated trastuzumab for HER2-positive gastrointestinal cancers

Major revision

Journal Requirements:

I revised my manuscript, and confirmed that my revised manuscript meets PLOS ONE's style requirements.

I cited our research group’s report (https://www.mdpi.com/2072-6694/11/5/636) to manuscript that was previously published. However, I can’t point out the overlapping text with the publication https://www.mdpi.com/1422-0067/25/6/3233. Although there may be some overlap in the methodologies, it is minimal.

I have no Supporting Information file.

There is no recommendation to cite specific previously published works.

Reviewer’s Comments to Author:

Reviewer: 1

This report relates to the use of a photosensitizing agent for treatment of GI cancer.

1) Line 34: if the selectivity of the photosensitizer is appropriate, there should be no damage to ‘normal mucosa’ because these cells should not be photosensitized.

Thank you for your comment. To the best of our knowledge, there is no photosensitizer that shows absolute and exclusive accumulation in tumor cells without any distribution to normal cells. Furthermore, while it is theoretically correct that super selective photosensitizers should not damage normal mucosa, drug distribution can be affected by inflammation, regenerative epithelium, and tumor heterogeneity in clinical cases. Therefore, improving selectivity remains an important clinical challenge. Tumor selectivity in photodynamic therapy remains a relative concept, and strategies to further enhance tumor-specific accumulation are still required.

2) If there is no selectivity for sites of neoplasia, a CO2 laser could be used for therapy.

Thank you for your comment. While CO₂ laser ablation may be feasible for superficial tumor removal, the extremely shallow penetration depth limits its usefulness for reducing advanced tumor volume in gastrointestinal cancers. Therefore, CO₂ laser therapy aimed at gastrointestinal tumor debulking is not commonly adopted in current clinical practice. In contrast, photodynamic approaches offer the potential to affect tumor tissue beyond the immediate surface.

3) It appears that the agent being used is thought to have insufficient targeting properties so it need to be conjugated to an antibody. There are many photosensitizing agents in use today that do not need to be conjugated to anything.

We appreciate your comment. Talaporfin sodium (TS) is currently the most widely used photosensitizer in clinical practice in Japan. Compared with the conventional agent porfimer sodium, TS exhibits enhanced tumor accumulation and a substantially shorter light-shielding period, which has led to its widespread clinical adoption in Japan. G-Ce6 is a novel photosensitizer that we synthesized and reported in 2018. In vitro photodynamic therapy using G-Ce6 demonstrated an anti-tumor effect that was approximately 9,000–34,000 times stronger than that of TS (These points were described in Reference 8. In addition, we have added an additional reference 11 (Osaki et al, Cancers, 2019.) related to G-Ce6 in the revised manuscript.). Taken together, these findings indicate that G-Ce6 is a highly potent photosensitizer with markedly enhanced cellular accumulation and photodynamic efficacy. The rationale for using an antibody-conjugated photosensitizer in this study was to introduce molecular targeting toward HER2-positive tumors. Our approach is intended as a complementary strategy for HER2-positive tumors, rather than a universal replacement for existing agents. Enhanced tumor accumulation compared with conventional agents provides several clinical advantages, including not only a reduction in damage to surrounding normal mucosa caused by scattered light, but also the feasibility of a shorter light-shielding period.

4) In vitro studies involved a 16 J/sq cm light dose which is very high for in vitro experiments. For most such studies, light doses in the range of 300-500 mJ/sq cm are used.

Thank you for your comment. We reviewed several publications on photodynamic therapy (PDT) using TS and found that, in in vitro experiments, irradiation is generally performed at the same order of magnitude as in our study, and none of the studies used irradiation doses in the mJ/cm² range. (7.5 J/cm2; Ezzaldeen et al, 2025, 10 J/cm2; Miki et al, 2013, 13 J/cm2; Akter et al, 2019, 0～10 J/cm2; Ichikawa et al, 2013). Therefore, we believe that this irradiation dose cannot be considered outside the standard range.

5) Details for the ROS assay indicate (line 120) that cells were irradiated, then exposed to DCFH-DA which is incorrect. DCFH-DA needs to be present during irradiation, since the ROS being formed are rapidly quenched. This error indicates that the authors are inexperienced in PDT technology.

Thank you for your comment. According to the protocol for H2DCFDA that we used (Invitrogen™), the recommended incubation time is 5–60 minutes. It is well known that it may cause dye–dye quenching when H2DCFDA is present in excess within cells. As you pointed out, we performed irradiation after co-incubating both the photosensitizers (G-Ce6 and G-Ce6-trastuzumab) and H2DCFDA for 60 minutes so that H2DCFDA would be present during irradiation; however, this approach did not yield satisfactory results. We consider that this may be due to the 60-minute incubation being insufficient for adequate cellular uptake of the photosensitizers.

An alternative approach—incubating cells with the photosensitizer first, washing, and then adding H2DCFDA in PBS before irradiation—was not performed, because the concentration of the photosensitizer used in this study (9.6n mol/L ; we apologize for the error in the initial version and have corrected it) was relatively low, and changing to PBS and incubating for longer time under PBS before irradiation was a concern as it could further dilute the photosensitizers.

Therefore, consistent with several previous reports (Huang et al, BMC cancer, 2019; Fatima et al, Journal of Advanced Research 2021), we administered H2DCFDA immediately after irradiation and evaluated ROS generation under these conditions.

6) What is the basis for the cell counting kit as an indicator of photokilling? The most unambiguous approach is to use a clonogenic assay which measures the ability of cells to form colonies. Anything else is only an approximation. Oleinick described a comparison between clonogenic assays and an alternative approach (Photochem Photobiol 2009) and found that the correlation was poor.

Thank you for your comment. We acknowledge that clonogenic assays are the gold standard for evaluating long-term reproductive cell death and are particularly appropriate for assessing cytotoxicity after prolonged drug exposure.

In this study, however, PDT involves a single light irradiation, and cytotoxic effects are typically evident within 24 hours after treatment. Therefore, for evaluating acute photokilling following a single PDT session, we consider the cell counting kit assay to be appropriate.

In contrast, clonogenic assays require long-term culture, during which additional confounding factors may influence colony formation. While clonogenic assays are valuable for long-term survival analysis, we believe that the cell counting kit assay is suitable for the aims of the present study.

7) Line 232 indicates a ‘specific wavelength of light’. All wavelengths of light are ‘specific’. What is needed is light at a wavelength that corresponds to an absorbance optimum of the photosensitizer, preferably one that can penetrate into tissues.

Thank you for your comment. We acknowledge that the term “specific” may cause misunderstanding. Accordingly, we have revised the expression to “optimal for the photosensitizer” to more accurately reflect our intention.

Summary: The ROS detection procedure should be carried out with the reagent present during irradiation. Any result that claims to detect photokilling should involve a clonogenic assay which detects the ability of cells to proliferate. It appears that the photosensitizer being used requires the presence of an antibody to provide adequate targeting. There are photosensitizers that do not have such a requirement. Does this agent represent a significant improvement over what is already available? It is true that the use of 654 nm light would represent an improvement over the 630 nm wavelength used with Photofrin.

Thank you for your valuable comments. We acknowledge that the presence of the ROS detection reagent during light irradiation is an important consideration. However, as described above, due to issues related to the concentration of the photosensitizer and the required incubation time, we addressed this concern by adding the ROS detection reagent H2DCFDA immediately after irradiation. In this study, our primary objective was to evaluate acute photokilling following a single PDT session using the cell counting kit assay. The newly developed photosensitizer G-Ce6-trastuzumab is designed to achieve high tumor accumulation through antibody-mediated targeting. The key features of this photosensitizer include the potential to reduce phototoxicity in non-tumor tissues by improving tumor specificity, as well as a reduction in the required drug dose. We consider these points to represent important advantages of this agent over existing photosensitizers.

Furthermore, the ability to use long-wavelength light at 654 nm offers a clear advantage in terms of improved tissue penetration compared with the 630 nm wavelength used for Photofrin, which we believe is of clinical relevance.

Taken together, compared with conventional photosensitizers, this novel photosensitizer G-Ce6-trastuzumab demonstrates clinical significance in terms of (i) enhanced tumor selectivity through targeting, (ii) reduced nonspecific phototoxicity, and (iii) improved tissue penetration enabled by long-wavelength excitation.

---

## [Editor Report · Decision Letter 2]

14 Jan 2026

PONE-D-25-57729R2

Anti-cancer effect of a novel photodynamic therapy using glucose-linked chlorin e6 conjugated trastuzumab for HER2-positive gastrointestinal cancers

PLOS One

Thank you for submitting your manuscript to PLOS One. After careful consideration, we have decided that your manuscript does not meet our criteria for publication and must therefore be rejected.

PLOS follows guidelines of the Committee on Publication Ethics (COPE) when concerns are raised about submitted and published work. Upon evaluation of the manuscript, we have concerns about the article’s adherence to PLOS policies and ethical publishing practices. As a result of these concerns, we cannot consider the manuscript for publication at this journal. This decision has been made on the basis of compliance with journal policy and is independent of any reviewer reports obtained.

Please note that due to the nature of the issues in this case and in line with PLOS policy on Manipulation of the Publication Process (https://journals.plos.org/plosone/s/ethical-publishing-practice#loc-manipulation-of-the-publication-process), we will not provide detailed information about the specific concerns underlying this decision.

I am sorry I do not have more positive news for you on this occasion.

Regards,

Xin Sun, PhD

Staff Editor

PLOS One
---

## [Author Response · Author response to Decision Letter 3]

3 Mar 2026

Manuscript ID: PONE-D-25-57729R2

Title: Anti-cancer effect of a novel photodynamic therapy using glucose-linked chlorin e6 conjugated trastuzumab for HER2-positive gastrointestinal cancers

Dear Editor and Reviewers,

We sincerely appreciate the opportunity to resubmit our revised manuscript and are grateful for the careful evaluation of our work. We have thoroughly revised the manuscript in response to all reviewer comments and have clarified methodological details and limitations to ensure scientific accuracy and transparency.

Below we provide a point-by-point response.

Reviewer #1

1. “Aging population” statement

We thank the reviewer for this important comment.

We agree that cancer can occur at any age. We have revised the Introduction to clarify that while cancer affects all age groups, the absolute number of cases increases with population aging due to cumulative genetic and environmental factors. We also clarified the clinical rationale for minimally invasive treatments in elderly populations.

The revised text has been incorporated accordingly.

2. Evidence for mucosal damage

We appreciate this insightful comment.

We have now added relevant preclinical and clinical references demonstrating that higher irradiation doses may cause damage to surrounding esophageal mucosa and contribute to post-PDT strictures. These include data from canine models and recent clinical reports.

The manuscript has been revised to include these references and contextual clarification.

3. Light dose used in vitro

We thank the reviewer for raising this concern.

We have re-examined the literature and confirmed that irradiation doses of similar magnitude have been reported in TS- and G-Ce6–based in vitro PDT studies. In addition, G-Ce6–based systems have occasionally required relatively high light doses.

We have clarified this point in the revised manuscript and included additional supporting references.

4. Use of PBS instead of culture medium during irradiation

We appreciate this important methodological question.

Phenol red in culture medium absorbs at wavelengths overlapping with PDT excitation spectra and may interfere with irradiation. Therefore, washing with PBS before irradiation is a widely accepted approach in short-duration in vitro PDT experiments.

We have clarified this rationale in the revised manuscript.

5. ROS measurement methodology

We sincerely thank the reviewer for this highly important comment.

We fully acknowledge that DCFH-DA ideally should be present during irradiation to capture short-lived ROS. However:

The photosensitizer concentration used (9.6 nmol/L) was relatively low.

Prolonged pre-irradiation incubation under PBS conditions risked dilution or leakage.

Co-incubation approaches did not produce reproducible results.

Therefore, H2DCFDA was administered immediately after irradiation, which primarily detects relatively long-lived ROS (e.g., peroxides).

Importantly, we have now:

Explicitly stated this limitation in the Discussion

Clarified that our method does not comprehensively quantify all ROS species

Modified the wording to avoid overinterpretation

We agree that this method does not provide a complete quantitative estimate of all ROS species generated during PDT.

6. Use of the term “viability” and lack of clonogenic assay

We thank the reviewer for this critical comment.

We fully agree that clonogenic assays represent the gold standard for assessing long-term reproductive cell death. However, our study aimed to evaluate short-term cytotoxic effects 24 hours after a single PDT exposure.

Therefore:

We used the CCK-8 assay (WST-8 based), which measures intracellular dehydrogenase activity and reflects metabolic activity proportional to surviving cell number.

We have replaced all instances of “viability” with “cell survival rate at 24 hours after PDT.”

We have clarified what CCK-8 measures.

We have explicitly acknowledged that clonogenic survival was not assessed and that slower cell death mechanisms were not evaluated.

We have carefully revised the manuscript to ensure terminological precision and avoid overstatement.

Reviewer Summary Comment

The reviewer expressed concern that without an appropriate ROS assay and clonogenic assay, the claims may not be fully supported.

We appreciate this important observation.

In response:

We have clarified the limitations of ROS detection.

We have corrected terminology regarding “viability.”

We have carefully revised the Discussion to ensure that conclusions are restricted to the data obtained.

We have avoided claims implying comprehensive ROS quantification or long-term reproductive death.

We believe the revised manuscript now accurately reflects the scope and limitations of the data presented.

We sincerely thank the reviewers for their constructive and scientifically rigorous comments. We believe that the revisions have substantially improved the clarity, transparency, and scientific precision of the manuscript.

Sincerely,

Mamoru Tanaka, M.D., Ph.D.

---

## [Editor Report · Decision Letter 3]

23 Mar 2026

Anti-cancer effect of a novel photodynamic therapy using glucose-linked chlorin e6 conjugated trastuzumab for HER2-positive gastrointestinal cancers

PONE-D-25-57729R3

Dear Dr. Tanaka,

We’re pleased to inform you that your manuscript has been judged scientifically suitable for publication and will be formally accepted for publication once it meets all outstanding technical requirements.

Kind regards,

Michael R Hamblin

Academic Editor

PLOS One
---

## [Editor Report · Acceptance letter]

PONE-D-25-57729R3

PLOS One

Dear Dr. Tanaka,

I'm pleased to inform you that your manuscript has been deemed suitable for publication in PLOS One. Congratulations! Your manuscript is now being handed over to our production team.

Kind regards,

on behalf of

Dr. Michael R Hamblin

Academic Editor

PLOS One